# *Brucella*: Reservoirs and Niches in Animals and Humans

**DOI:** 10.3390/pathogens10020186

**Published:** 2021-02-09

**Authors:** Gabriela González-Espinoza, Vilma Arce-Gorvel, Sylvie Mémet, Jean-Pierre Gorvel

**Affiliations:** Aix Marseille Univ, CNRS, INSERM, CIML, Centre d’Immunologie de Marseille-Luminy, 13288 Marseille, France; gonzalez@ciml.univ-mrs.fr (G.G.-E.); arce@ciml.univ-mrs.fr (V.A.-G.)

**Keywords:** *Brucella*, replication niche, reservoir, persistence, survival, chronic infection

## Abstract

*Brucella* is an intracellular bacterium that causes abortion, reproduction failure in livestock and leads to a debilitating flu-like illness with serious chronic complications if untreated in humans. As a successful intracellular pathogen, *Brucella* has developed strategies to avoid recognition by the immune system of the host and promote its survival and replication. In vivo, Brucellae reside mostly within phagocytes and other cells including trophoblasts, where they establish a preferred replicative niche inside the endoplasmic reticulum. This process is central as it gives *Brucella* the ability to maintain replicating-surviving cycles for long periods of time, even at low bacterial numbers, in its cellular niches. In this review, we propose that *Brucella* takes advantage of the environment provided by the cellular niches in which it resides to generate reservoirs and disseminate to other organs. We will discuss how the favored cellular niches for *Brucella* infection in the host give rise to anatomical reservoirs that may lead to chronic infections or persistence in asymptomatic subjects, and which may be considered as a threat for further contamination. A special emphasis will be put on bone marrow, lymph nodes, reproductive and for the first time adipose tissues, as well as wildlife reservoirs.

## 1. Introduction

*Brucella* is a Gram-negative facultative intracellular coccobacillus, responsible for brucellosis, a worldwide zoonotic disease that affects livestock, wildlife, and humans. Brucellosis remains endemic in many parts of the world, including the Middle East, Africa, Latin America, Central Asia, and several regions of the Mediterranean [1,2,3,4]. *Brucella melitensis*, *Brucella abortus,* and *Brucella suis* are the most important members of the genus because they are responsible for the human disease [5,6] and for significant economic losses in livestock [7,8], while *Brucella canis*, *Brucella neotomae*, and *Brucella ovis* display less or non-zoonotic potential [9]. Recently, many other species have been described in aquatic mammals (like *Brucella pinnipedialis* and *Brucella ceti*) and other wildlife such as *Brucella papionis*, *Brucella microti*, *Brucella inopinata,* and *Brucella vulpis* [10,11,12,13,14,15,16].

### 1.1. Brucellosis, a Zoonosis

Humans are accidental hosts for *Brucella* and are mainly infected through direct contact with infected animals, inhalation of airborne agents, or by ingestion of contaminated dairy products [17]. Human-to-human transmission may occur by organ transplantation, blood transfusion, or vertical transmission via breastfeeding [18,19,20,21,22]. Human brucellosis is at the origin of many symptoms namely undulating fever, malaise, fatigue, and anorexia. If untreated, it may progress into a chronic phase, characterized by the appearance of severe complications like endocarditis, orchitis, spondylitis, osteomyelitis, arthritis, meningoencephalitis, and recurring febrile conditions [23,24,25,26].

In domestic animals, such as cattle, sheep, goats, and swine, major consequences include abortion and metritis in females, and orchiepididymitis and infertility in males [27], resulting in reduced fertility and a significant decline in milk production [28]. Animal brucellosis is highly contagious for both animals and humans, and cross-species transmission of certain *Brucella* spp. exists [29]. Natural infection occurs by direct contact with infected animals or their secretions [30], like aborted fetuses and fetal membranes that contain large amounts of the bacteria [31].

### 1.2. Infection and Dissemination

*Brucella* organisms enter into their host through the mucosal membranes of the respiratory and digestive tracts [32]. Once inside, local professional phagocytes such as macrophages, dendritic cells, and neutrophils internalize the bacteria and move to the closest draining lymph nodes following the normal sampling of the immune system. This leads to subsequent dissemination to the different organs of the reticuloendothelial system, including lungs, spleen, liver, and bone marrow [33]. In pregnant animals, *Brucella* displays a strong tropism for placental trophoblasts [34,35,36] and also for mammary glands [37], in which it replicates extensively causing placentitis and abortion in the last trimester of pregnancy in ruminants [33]. In humans, brucellosis is a systemic infection and any organ can become infected, albeit with some predilection for joints and liver, and at lower levels for the brain and heart [38].

### 1.3. Acute and Chronic Infections

Human brucellosis presents a broad range of clinical manifestations from asymptomatic to severe disease. When symptoms are present, the incubation period is 1–4 weeks, but sometimes it takes several months with or without signs and symptoms. According to the duration of the symptoms, human brucellosis is classified into three different phases: acute (initial, 2 months), sub-acute (2–12 months), and chronic (more than 12 months) [39,40]. When focusing on the chronic phase, patients fall into three more categories: relapse (with fever or high IgG antibody titers after antibiotics therapy), chronic localized infection (recurrence of infectious foci and intermittent fever for long periods of time), and delayed convalescence (persistence of some symptoms without fever or other objective signs) [41].

In animals, brucellosis comprises three phases: the onset of the infection or incubation period, when *Brucella* invades the host without any clinical symptoms; the acute phase, when the bacteria replicate actively and infection remains unapparent in most cases or first pathological signs arise; and the chronic phase, during which bacterial loads reach a plateau before decreasing and sporadic clinical symptoms become visible, the infection localizing in sexually mature animals in the reproductive system to produce epididymitis or orchitis in males, and placentitis and abortion in females. Susceptibility to infection in females increases during the late pregnancy stages (third trimester of gestation) [31,42]. There is no pyrexia like in humans and death is rare; infection is self-limiting most of the time and does not involve other systemic lesions, but for *Brucella suis* infection in swine. Clinical symptoms for the latter include spondylitis of the lumbar and sacral regions, arthritis, paralysis of hind limbs, lameness, and abscesses in tissues [42]. In any case, chronicity relies on the continuous shedding of *Brucella* from the mammary gland or reproductive organ secretions for a protracted period. Therefore, infertility, repetitive abortions, and premature stillborn ensure *Brucella*’s permanence in the environment and liability for dissemination.

The long incubation period and the absence of obvious clinical signs in infected animals and patients correlate with low activation of innate immunity [43]. *Brucella* has indeed modified its pathogen-associated molecular patterns (PAMPs) into less detectable molecules, opening a permissive immunological window to spread stealthily throughout the reticuloendothelial system and find its target host cells [44]. Then, invasion of target cells occurs thanks to specific molecular determinants that drive not only ingress [45] but also resistance to intracellular killing allowing *Brucella* to reach its intracellular replicative niche within professional and non-professional phagocytes [46]. Even if innate immunity first restrains *Brucella* proliferation, a Th1 response with IFNγ and IL-12 production is absolutely required to eradicate it, leading either to complete clearance of the bacteria or to a chronic infection, which arises from the ability of *Brucella* to persist undetected for prolonged periods of time within its host reservoirs [44,47].

We propose that the intracellular replication niche conditions the basis for the establishment of the reservoirs in which *Brucella* persists inside its host. What is currently known of the suitable niches for *Brucella* infection in its various hosts and of the reservoirs of *Brucella* that support chronic infection?

## 2. Niche and Reservoir

### 2.1. Definitions

The concepts of niche and reservoir may be extrapolated to the infection context. In general, Niche is referred to as the “interrelationship of a species and its relational position in a particular ecosystem including the relationship of the species with the components of the ecosystem itself” [48]. As such, the niche may be influenced by all the factors included in its ecosystem and the niche of a species in a particular ecosystem helps setting up the features of its environment, as the latter ones are crucial for its survival [48]. A specific niche is thus defined as: (i) the specific area where an organism inhabits, (ii) the role or function of an organism or species in an ecosystem, and (iii) the interrelationship of a species with all the factors affecting it. The niche of an organism depicts how it lives and survives as a part of its environment. *Brucella* is a pathogenic genus exceedingly well-adapted to its hosts, which does not survive for extended periods of time in open conditions. This is why it has been called a facultative extracellular intracellular parasite [49]. It also means that *Brucella* has a niche in the intracellular environment of host cells that is specific in a specific cell of the host. This environment sustains extensive replication, in order to facilitate bacterial expansion and subsequent transmission to new host cells, frequently achieved for example, through the heavily infected tissues like the aborted fetus [49].

In the case of Reservoir, the term is referred to an ecologic species that maintain live circulating organisms through the ecosystem over time. For instance, within the niche from a host reservoir, *Brucella* can remain at a low replication rate for a long time, and under favorable conditions, egress to infect other cells and start new replicative cycles. In addition, *Brucella* infections in humans perdure in the ecosystem due to the lack of control of the infection in natural hosts [6,50]. Lymph nodes, spleen, lungs, and the reproductive organs, including placenta, testicular and mammary glands, are well-known target organs for *Brucella* infection [27]. Some predilection for joint articulations has also been reported in human brucellosis [23], meaning that all these organs allow *Brucella* replication and that some may contribute to persistence.

### 2.2. Intracellular Niche

*Brucella* replicates extensively in the endoplasmic reticulum (ER) compartment within host cells. The mechanisms of entry of the bacterium are still elusive but involve lipid raft-, adhesin- and opsonin-dependent processes [51,52,53,54]. After internalization, *Brucella* transits inside the cell engulfed in a phagosome, and multiple virulence factors help the bacteria evade the phagocytic pathway by restricting fusion of the *Brucella* containing vacuole (BCV) with a lysosome. These factors include the cyclic beta-1,2-glucan that operates most probably via cholesterol release [55], the two-component system BvrR/BvrS [56], SepA, which proceeds by excluding the LAMP1 lysosomal protein and preventing the maturation of an active lysosome [57], RicA that regulates vesicle trafficking [58,59] and other possible proteins secreted by the type IV secretion system (T4SS), encoded by the *virB* operon [60,61]. In vitro experiments using macrophage cell lines have indeed shown that the T4SS is required for maturation of the BCV into an ER-like compartment [62]. *Brucella* strains lacking a functional T4SS are unable to escape the fusion with lysosomes, and therefore, highly attenuated in mice and in their natural hosts [63]. Once *Brucella* has impaired phagosome-lysosomal fusion, it replicates in the ER compartment, its replicative niche. Following replication, BCVs interact with host autophagic proteins Beclin1, ULK1, Atg14, and the IRE1α-UPR signaling axis for bacterial egress and the start of replication cycles within newly infected cells [64,65,66,67].

Why does the ER make a good intracellular niche for *Brucella*? The ER is a critical intracellular organelle that not only synthesizes cellular molecules (proteins, lipids, carbohydrates, etc.) but also regulates the transport of the newly synthesized proteins in the exocytic, endocytic and phagosomal pathways. As such, the association of *Brucella* with the host cell ER, like a few successful intracellular pathogens, is expected to be highly beneficial from a nutrient acquisition perspective. Taking advantage of the biosynthetic routes of the host cell, substantial levels of metabolites and nutrients on a local supply base fulfill the complex nutritional requirements of *Brucella* and provide optimal bacterial growth at minimum cost [27,68]. Furthermore, when considering the immune response, localization of *Brucella* in the ER provides an excellent strategy to hide from detection by the immune system and to limit exposure to the cytosolic immune surveillance pathways by avoiding lysosomal fusion for instance. Moreover, the fact that the MHC I peptide loading complex resides in the ER of the two immune cells where *Brucella* replicates, the dendritic cells (DC) and the macrophages, is likely not meaningless. It may suggest a yet unidentified regulatory role of *Brucella* at the level of the setting up of the cross-presentation within the ER and explain its predilection for such cell types.

Indeed, the preference of *Brucella* to replicate within the ER is mostly restricted to phagocytic cells, professional ones as macrophages [69] and DC [70], and non-professional ones, such as placental trophoblasts of pregnant ruminants [35], and fibroblasts [71] or cell lines (Hela) [72]. Even though other cells, including neutrophils [73], lymphocytes [74], and erythrocytes [75], are infectable by *Brucella*, there is no efficient replication inside, and their function is more associated with bacterial dispersion, conferring a regulatory role of these cells in persistence. Of note, if in most cell types *Brucella* replicates within an ER-derived compartment, in extravillous HLA-G+ trophoblasts, *B. abortus*, and *B. suis* fail to reach the “normal” ER-derived niche, in contrast to *B. melitensis*, and replicate within single-membrane acidic lysosomal membrane-associated protein 1 (LAMP1)-positive inclusions [76].

## 3. Gold Organs in Brucellosis

The “gold organs” for nesting *Brucella,* in which *Brucella* replicates in cells of the reticular endothelial system, include the spleen, lymph nodes, liver, bone marrow, epididymis, and placenta.

### 3.1. The Reticuloendothelial System

The reticuloendothelial system was originally described in 1924 by K. Aschoff as a group of cells able to incorporate vital dyes from the circulation, “reticulo” referring to their propensity to form a network or reticulum by their cytoplasmic extensions and “endothelial” referring to their vicinity to the endothelium. In 1969, a group of pathologists proposed another term, the monocyte phagocyte system (MPS) [77]. Nowadays the reticuloendothelial system or MPS embraces a family of cells that include committed precursors in the bone marrow, circulating blood monocytes, tissue macrophages, and DC in almost every organ in the body [78].

*Brucella* has a predilection for organs rich in reticuloendothelial cells (including spleen, liver, bone marrow, and lymph nodes) and is able to replicate successfully in any of them. Intracellular replication is directly linked to *Brucella* pathogenicity and it is not a coincidence that in humans, the most frequent clinical features of brucellosis are an enlarged liver in 65% of the cases, splenomegaly in 52% of the cases (from 40 cases), and lymphadenopathies in children [32,79]. Even in the chicken embryo model, replication of *B. abortus* detected within the rough ER of mesenchymal, mesothelial, and yolk endodermal cells, spreads to all tissues, with the liver and spleen being the most severely infected [80].

In tissues, the typical histopathological response to *Brucella* infection is a granulomatous inflammation, which contains representative members of the MPS, including macrophages with an epithelioid shape, i.e., with an increased amount of cytoplasm. Examination of biopsies from humans and livestock animals reveals granulomas in the liver, spleen, bone marrow, and other tissues [79,80,81,82]. As such, the initial replication niche of *Brucella* serves as a platform to establish a chronic infection. *Brucella* infected animals develop granulomatous inflammatory lesions in lymphoid tissues, including the supramammary lymph nodes, reproductive organs, notably the udder, and sometimes joints and synovial membranes. Those granulomas and their intratissular location are responsible for the chronicity of the disease, which can last for months or years [81,83] and in that respect, resemble the granulomas extensively studied in tuberculosis. In fact, in the absence of antibiotic treatment in the acute phase, *Brucella* is able to persist for months without causing significant morbidity or mortality. In the acute phase of infection in a resistant mouse model, the C57BL/6 mice, the formation of granuloma (comprising NOSII+ monocyte-derived inflammatory DC, T cells, and granulocytes) is mediated by MyD88, IL-12, and IFNγ and essential for the control of the bacteria [81,83]. However, these granulomas were not detected in a susceptible murine model of infection, the BALB/c mice, at that stage [81,83]. In *B. melitensis* acutely infected livers, discrete pyogranulomatous inflammatory areas, characterized by a similar influx of neutrophils, macrophages, and monocyte-derived DC, were detected amongst normal hepatocytes in both mouse models [81,83]. At the chronic phase, infected livers displayed well established demarcated infiltration areas of macrophages, lymphocytes, and neutrophils [81]. In chronic granulomas, the presence of lymphocytes is thought to reflect the former activation of the immune system, whereas recruitment of neutrophils suggests that live *Brucella* is still present. The fact that the granuloma areas were typically found surrounding or associated with liver portal tracts and that neutrophils may function as vehicles for dispersion, according to the Trojan horse model [84], supports a dynamic role of granulomas in the development of *Brucella* chronicity. Remarkably, granulomas provide a rich nutrient source, as shown for the dormant non-replicative *Mycobacterium bacilli* that internalize inside the granuloma, lipids from foamy macrophage lipid droplets [85].

### 3.2. Genital-Reproductive Organs: Placenta and Epididymis

*Brucella* has a pronounced tropism for genital organs in its natural hosts, placenta in females, and epididymis in males. The placenta is one of the paradisiac organs in terms of replication, containing up to 10^14^ Brucellae in the cow [86,87]. This particular environment allows high replication rates, leading consequently to abortion, the most common clinical feature of brucellosis in livestock. As the main route of infection in these farm animals is aborted fetuses, this seems to be a very efficient strategy to spread *Brucella* progeny to new hosts.

Some common properties in these reproductive organs have shed light on *Brucella’s* tropism. Firstly, high concentrations of erythritol are present in uterine, epididymal, and fetal tissues from ruminants [87,88,89,90]. Why is this important? Erythritol has been shown to be the preferred carbon/energy source for *Brucella* spp., promoting their massive growth [91]. In addition, the ruminant placenta produces progesterone, which further enhances in vitro *B. abortus* growth [92]. However, *B. abortus* vaccine strain S19 is not stimulated by erythritol [93,94], although it is capable of causing genital infection and abortion [95]. This suggests the existence of other trophic factors. Indeed, the dominance of fructose over glucose takes place in the placenta of cows, sows, ewes, and to a lesser extent in that of other animals [91,96,97]. The same preference applies to the epididymis, seminal fluids, and oviducts of several mammals [91]. As such, both organs play a trophic role and provide effective sources of carbon, nitrogen, and energy for *Brucella* spp. [49,91].

Secondly, the immune-privileged status of the testis and semen, and local immunosuppression at the feto-maternal interface in the placenta might also account for *Brucella* tropism [91].

Thirdly, *Brucella* preferentially replicates within trophoblasts, highly metabolically active cells that adjust their production of proteins and steroids throughout gestation. Intracellular *Brucella* likely induces the synthesis of steroids and modifies the metabolism of prostaglandin precursors, such as arachidonic acid, which together with the COX-2 enzyme are essential for *Brucella* lymph node persistence and subversion of the immune response [98].

Finally, the high hydrophobicity of the outer-membrane of *Brucella* together with its propensity to replicate within the ER [35,99], may represent an evolutionary adaptation for using hydrophobic substances available within this sub-cellular compartment in trophoblasts [49].

In humans, the genital tropism holds true as *Brucella* induces epididymorchitis [100] and may infect the placenta, even if abortion is very uncommon [76,101].

Therefore, both the localization and abundant multiplication in the reproductive tract of animals is crucial in the biology of this pathogen.

## 4. Reservoirs

### 4.1. Bone Marrow

The presence of *Brucella* organisms in humans has been highlighted in the bone marrow in both the acute and chronic phases. In the mouse, the infection remains sequestered within bone marrow cells for a prolonged period of time (until more than 3 months), without significant changes in the bacterial load [102]. For this reason, the bone marrow has recently been proposed as a reservoir [102].

#### 4.1.1. Bone Marrow in the Mouse Model

Murine models have validated bone marrow as an intermittent colonized organ by *Brucella*. Indeed 3 weeks p.i., *Brucella melitensis* has been detected in multiple sites of the murine axial skeleton by in vivo imaging, and immunohistochemistry confirmed its presence in bones, particularly in the lower spine vertebrae, where it preferentially located in a small subset of IBA-1+ monocytes [103]. Similarly, *B. canis* bacterial loads increased in mouse bone marrow from 9 weeks p.i. onward till 12 weeks, a time which coincides with the persistence and chronicity of the infection [104]. Recently, Gutiérrez et al. observed that *Brucella abortus* burden remained constant in bone marrow for up to 168 days p.i., including the acute, chronic, and chronic declining phase in the murine model [102]. However, histopathological alterations varied accordingly to the stage of infection. A granulomatous inflammation, accompanied by augmented numbers of myeloid granulocyte-monocytes progenitors (GMP), granulocytes, and CD4+ lymphocytes, was more severe and diffuse during the acute phase than that of the multifocal chronic phase [102].

The vast number of granulomas in the bone marrow and most importantly their permanence indicates the difficulty for immune cells to eliminate *B. abortus* in such an environment. Interestingly, the cells harboring *Brucella* in higher proportion were granulocytes, monocytes, and GMP progenitors. The fact that monocytes are constantly infected and controlled by *Brucella* strongly suggests that bone marrow is a proper reservoir for persistence in brucellosis.

Moreover, the possibility that *Brucella* infects progenitor stem cells, which lack developed phagocytic machinery, is striking. Recently, it has been shown that hematopoietic stem cells (HSC) sense pathogens, eliciting enhanced myeloid commitment to promoting pathogen clearance of *E. coli* and *Salmonella* Typhimurium [105,106]. In the case of *Brucella*, an increased transient myeloid commitment is triggered by the interaction of *Brucella* Omp25 and host SLAMF1 in the bone marrow, favoring bacterium survival [107]. This may be one of the multiple strategies developed by *Brucella* to promote chronic infection and drive bacterial dissemination.

#### 4.1.2. Bone Marrow in Humans

Bone marrow infection by *Brucella* also occurs in humans in both acute and chronic phases. Usually, a definitive diagnosis of brucellosis is made by culturing *Brucella* from body fluids or tissues [108,109]. The bone marrow has been a recommended tissue to investigate in suspicious cases of brucellosis when the blood culture test is negative. Gotuzzo et al. [110] determined that out of fifty patients, 92% of *Brucella* yielded in bone marrow versus 70% in blood. As mentioned above, granulomas are formed as a reaction to particulate or indigestible agents that persist in tissues for long periods. Additionally, 25% of *Brucella*-infected bone marrows show granulomas together with other pathological changes such as hypercellularity (73% of all cases) or hemophagocytosis (31%) [108,111]. These conglomerates of macrophages, which try to destroy the microbial agent and interact at the same time with lymphocytes might explain rare cases of pancytopenia that are observed in some patients with brucellosis [112].

Human to human brucellosis transmission is extremely rare. In fact, the transmission is due to external factors that promote the transfer of *Brucella*. Bone marrow transplantation facilitates such a transfer, as a concentration of bone marrow cells that are sheltering *Brucella* are transferred to a new host. This means that even in the case of an asymptomatic patient where *Brucella* is present, hidden within bone marrow cells, it will then replicate in the acute phase, or remain there establishing a chronic infection in the recipient host [113,114].

Bones appear to be support structures metabolically inert and resistant to infection by pathogens. However *Brucella* has a tropism for such location and osteoarticular brucellosis is the most common complication in *Brucella*-infected humans (40% from total complications) involving sacroiliitis, spondylitis, and peripheral arthritis [38]. Importantly, osteoarticular lesions are also reported in natural hosts, as infected cattle may exhibit bursitis, arthritis, and hygromas [115,116]. Previous studies have demonstrated the ability of *B. abortus* to invade and replicate within osteoblasts (bone-forming cells), osteocytes (bone-resorbing cells), osteoclasts (multinucleated giant formed by monocyte fusion) [117,118], and in the ER of primary human synoviocytes [119]. Remarkably, osteoclasts originate from the same myeloid precursor cells that give rise to macrophages and myeloid DC and many of the soluble factors (cytokines and growth factors) of immune cells that are regulated by *Brucella*, may regulate the activities of osteoblasts and osteoclasts. A common feature of patients with osteoarticular brucellosis is the presence of leukocyte infiltrates (including monocytes and neutrophils) in the synovial fluid of the joints [120]. The fact that *Brucella* inhibits apoptosis in some of those cells or uses them as a vector to spread to other organs, suggests that its osteoarticular location acts as a reservoir of bacteria to progress towards chronicity.

#### 4.1.3. Bone Marrow Environment

Blood cell formation or hematopoiesis is constantly occurring in the bone marrow, starting from self-replicating-pluripotent HSC, which give rise to multipotent progenitors and further on to lineage-committed cells, such as common lymphoid progenitors (CLP) or common myeloid progenitors (CMP). This sequence of events is tightly regulated by key cytokines and growth factors. The need for such a regulated process suggests the existence of a proper nursing environment called “HSC niche” provided by non-hematopoietic cells and by the undefined architecture of the bone marrow if compared to the spleen or lymph node. Cells forming the niche of HSC have to stay close to the progenitors, produce growth and maintenance of soluble factors, sense signals, and respond to the surrounding environment. This group is made up of endothelial, mesenchymal stem cells, macrophages, DC, granulocytes, megakaryocytes, and lymphocytes. In the HSC niche, amongst others, neutrophils modulate endothelium to produce vascular and HSC regeneration, while DCs control vascularity permeabilization processes [121]. Macrophages physically retain the HSC in the bone marrow and regulate HSC fate by fine-tuning the expression of pro-hematopoietic factors in other cells [122]. Moreover, macrophages via activation of cholesterol sensing LXR receptors eat apoptotic neutrophils that return to the bone marrow to be cleared and allow hematopoietic progenitor release into the blood circulation [123]. Memory CD8+ T lymphocytes and CD4+ Tregs are also localized in the bone marrow and protect HSC and downstream progenitors from various types of stress including inflammation [124].

The bone marrow environment is not only beneficial for HSC, pathogens have also taken advantage of those favorable conditions. *Mycobaterium tuberculosis* is found in mesenchymal cells and HSC of the bone marrow [125,126], which can also host *Listeria monocytogenes.* However, *Listeria* persists rarely in the bone marrow of humans and mouse bones, from where it ultimately seeds the central nervous system [127]. As mentioned before, *Brucella* is able to infect GMP, macrophages, and granulocytes. *Brucella* bone marrow infection is a successful strategy for the following reasons.

(i)Colonization of an immune-privileged organ that lacks intracellular antibacterial mechanisms in early-stage stem cells is easier.(ii)Expression of drug efflux pumps by bone marrow cells facilitates resistance to antibiotic therapy.(iii)Cells from bone marrow are highly motile and reside in close proximity to arterial vessels thus offering a wide distribution system for dissemination of the pathogen to target organs.(iv)*Brucella*’s control of HSC release might explain the splenomegaly observed in human cases as a result of the signal triggered in the bone marrow niche by the apoptotic infected neutrophils to the macrophages, which promotes ensuing cycles of HSC release and extramedullary hematopoiesis.(v)All *Brucella* preferential target cells differentiate from progenitor stem cells in the bone marrow.

Moreover, the fact that *Brucella* is transmittable by bone marrow transplantation in humans and adoptive transfer experiments in the mouse [107,128], indicates that *Brucella* is able to infect and persist for prolonged periods of time in bone marrow progenitors. It also means that the differentiated infected cells might derive from a GMP, which was already infected and recolonized target tissues in relapse or chronic cases.

### 4.2. Lymph Nodes

Lymph nodes (LNs) are essential compartments of the immune system, composed of highly organized dynamic leukocyte aggregates, where pathogen defense and adaptive immunity take place [129]. A tightly controlled balance of responses upon challenge results in the induction of either tolerance or immunity [130]. LNs filter fluids from the lymphatic vessels that form an extensive network connecting one LN to another. As such, this network provides conduits for the trafficking of DCs, neutrophils, macrophages, and T cells along the process of immune activation after capturing, transportation, and presentation of antigens in regional lymph nodes [131].

Some bacterial pathogens have exploited the lymphatic system for host colonization in LNs and systemic dissemination [132]. Lymphadenopathy is one of the most common signs of brucellosis in humans, present in approximately more than one-third of cases [4,32,133]. As an example, out of 307 children diagnosed with brucellosis, 112 (36%) exhibited lymphadenopathy [32]. LNs most commonly affected are cervical and axillary LNs due to their proximity to the oral route of infection, the natural route of infection for *Brucella* in humans.

In ruminants, *B. abortus* and *B. melitensis* have a marked affinity for mammary glands and reproductive organs together with the supramammary and genital LNs [134,135,136]. In goats, about two-thirds of infections acquired naturally during pregnancy lead to infection of the udder and excretion of the bacteria in the milk during successive lactations [137]. In contrast, in sheep, excretion of bacteria in the milk does not last more than two months generally, continuing for up to 140 or 180 days exceptionally [137].

In camels, *B. abortus* and *B. melitensis* have been isolated from LNs, and other organs or compounds like milk, aborted fetus, and placenta [138,139].

The constant or intermittent shedding of *Brucella* in the milk and genital secretions is held by the colonization of LNs and the subsequent spreading through lymphatics. Hence, *Brucellae* have been detected in macrophages [32] and neutrophils in lymphatic draining sites of inoculation or in mammary gland LNs [136]. Since LN leukocytes eventually enter the blood, transportation of infected phagocytes in the lymphatic network might disseminate bacterial organisms throughout the host. In fact, the spleen and the iliac, mammary, and prefemoral LNs are the most reliable samples for isolation purposes in necropsied animals thus proving efficient dissemination [140]. The persistent infection of LNs leads to a constant or intermittent shedding of *Brucella* in the system providing a source of persistent infection in the host and for other animals or humans. Indeed, close contacts between farmers and pastoralists with domesticated animals and the frequent consumption of fresh unpasteurized dairy products increase the maintenance of *Brucella* and the risk of brucellosis in rural and pastoral areas.

### 4.3. Adipose Tissue

Microorganisms show incredible diversity with respect to which environment they preferentially colonize or invade. Recently, the emergence of new investigation techniques has allowed the analysis of the tropism of several microbes in yet unexamined organs. This is the case for adipose tissue. Adipose tissue is no longer studied as inert lipid storage but as a central regulator of energy homeostasis and immunity. In the past years, fat-associated lymphoid clusters (FALCs) have received much attention [141]. These structures are quite peculiar because contrary to secondary lymphoid organs, FALCs lack both a surrounding capsule and structured compartmentalization of cells. In contrast, they are in direct contact with adipocytes at mucosal surfaces, including omental, mesenteric, mediastinal, gonadal, and pericardial fat [141]. Adipose tissue comprises a vast range of cellular and non-cellular components that support a large network of cells. These include fibroblasts, preadipocytes, cells with mesenchymal and hematopoietic stem cell capacity together with myeloid cells (macrophages, neutrophils, etc.), lymphocytes (T cells, B cells, innate lymphoid cells (ILCs)), eosinophils, mast cells, and NK cells [142]. These immune cells mostly aggregate in clusters or are scattered around them.

The presence of myeloid cell precursors and mature macrophages in the FALC milieu suggests that the lymphoid clusters form permissive microenvironments, where progenitor cells may proliferate locally to generate free macrophages within the cavity where they are located. Another feature of FALCs is that they are rich in vascularization; they are always closely associated with both blood and lymphatic vessels. In terms of response to pathogens, a rapid formation of acute or chronic inflammation occurs in FALCs, suggesting a prominent role of their clusters in the formation of local immune responses.

Many microorganisms, from viruses like HIV and SIV, to bacteria, like *Mycobacterium tuberculosis*, *Ricketssia prowazekii*, *Coxiella burnetti,* and parasites, such as *Plasmodium berghei*, *Trypanozoma cruzi,* and *Trypanozoma brucei*, have been shown to infect adipocytes from humans or mice [143,144,145,146,147,148]. Their pattern of infection varies according to each pathogen. Some of them establish an intracellular infection specifically within the adipocyte, whereas others remain close to or surround adipocytes and vasculature, or alternatively infect non-fat cells like macrophages.

Not so long ago, nothing was known about *Brucella* and adipocytes or fat tissues. One report in 2018 described the presence of *B. canis* in fat cells of the gastro-splenic ligament, next to lipid droplets and precisely where ER is located, in naturally infected fetuses and neonates [149]. More recently, *B. abortus* has been shown to replicate in a murine fibroblastic derived adipocyte-like cell line, the 3T3-L1 cells, and in its differentiated adipocyte derivative, albeit with less efficiency [150]. However, in this system, bacterial loads start to decline steadily from 3 days p.i. onwards, suggesting that this cell type may not serve as a long-term cellular niche for *Brucella* per se.

Based on these new findings, we would like to propose the adipose tissue in its whole as another singular reservoir for *Brucella* and speculate on the benefits this furtive bacterium might gain from such a location.

(i)Fat tissues are enriched in immune cells as aforementioned. With respect to macrophage populations, M2 polarized ones together with other immune cells recruited locally, like neutrophils, monocytes, and DC, might help to maintain *Brucella* survival and take control of the immune response during infection as suggested by the anti-inflammatory polarization of adipose tissue macrophages potentiated by chronic *T. cruzi* infection [145].(ii)The majority of drugs are lipophilic compounds, meaning that efficient distribution within the adipose tissue is hampered, as illustrated by some HIV treatment failures. In human brucellosis, 4% of cases undergo to chronic phase [151]. This incidence may rise because of delayed administration of antibiotics or its inefficient delivery to specific organs, such as fat tissue, in the acute phase of infection.(iii)The localization of fat depots in the perigonadal region of rodents, humans, and ruminants [152] and the preference of *Brucella* to infect epididymis might not just be a simple coincidence and reflect an essential role of adipose tissue in the persistence of this bacteria in reproductive organs. It also suggests that fat tissue provides *Brucella* with a proper environment, i.e., a rich source of nutrients.(iv)FALCs respond dynamically to stimuli. They expand in size and numbers in response to acute or chronic peritoneal insult [153]. More specifically, an increase of B1 cells, macrophages, and neutrophils recruited via the high endothelial venules has been described in omental FALCs upon injection of *E. coli* LPS or infection [154]. FALCs also support a unique population of CD4+ regulatory cells producers of IL-10 called visceral adipocyte tissue-associated (VAT) Tregs, mostly studied in large fat depots like epididymal fat [155] but also present in the omentum, where it likely regulates local immune responses [156]. Given the essential role of IL-10 in promoting *B. abortus* persistence and pathology [157] and the balance between activation and regulation of immune state that exists in adipocyte tissue, it is conceivable that pathogens, and *Brucella,* in particular, find a golden reservoir here, in which regulatory cells would control the local immune response and create a tolerogenic environment for progression to chronicity.

## 5. Reservoirs in Wildlife

In addition to the anatomic reservoirs described above, it is imperative to also consider wildlife host reservoirs. The control of brucellosis in humans depends on the control of disease in livestock. However, the creation of new bridges between livestock and wildlife due to human activity is one of the most important factors in disease transmission. As an example, brucellosis has been eradicated in domestic ruminants from most European countries and wild ruminants were not reckoned important hitherto reservoirs. This view changed recently after the notification of two humans cases and the re-emergence of *B. melitensis* in a dairy cattle farm [158], suggesting a possible implication of wildlife as *B. melitensis* infection was identified in a French population of Alpine ibex (*Capra ibex*) [159]. Interestingly, among the 88 seropositive Alpine ibex tested, 58% showed at least one isolation of *B. melitensis* from a urogenital organ (testes, genital tract, urine, or bladder) or a lymph node from the pelvic area (supramammary, internal iliac, and inguinal LNs), meaning that active infection in the pelvic area is at risk of shedding *Brucella* [159]. Moreover, microbiome analysis of bats’ guano in India found *B. melitensis* affiliated sequences [160], and *Brucella* sequences were detected in the spleen of two different species of bats from Georgia that were coinfected with *Bartonella* and *Leptospira* [161]. These data indicate that bats may serve as a wildlife reservoir of *Brucella* for grazing goats and sheep.

Regarding other classical *Brucella* species, infections that are recognized as sustainable in wildlife are *B. abortus* in buffalo (*Syncerus caffer*) [162] and bison (*Bison bison*) [163], *B. suis biovar* 2 in wild boar (*Sus Scrofa*), and European hare (*Lepus europeaus*) [164], *B. suis* biovar 4 in reindeer (*Rangifer tarandus*) [165], *Brucella ceti* in cetaceans (*Cetacea*) [166], *Brucella microti* in voles (*Microtus arvalis*) [167] and red fox (*Vulpes vulpes*) [168].

Other species genetically related to an atypical group within the genus have been described in different host species: *Brucella microti*-like in marsh frogs (*Pelophylax ridibundus*) [11]; *Brucella vulpis* in the red fox (*V. vulpes*) [13]; *Brucella inopinata* in White’s and Denny’s tree frogs, (*Ranoidea caerulea* and *Zhangixalus dennysi*) [169] and humans; *Brucella papionis* in baboons (*Papio* spp.) [15], without mentioning *Brucella* strains isolates from lungworms in porpoise (*Phocoena phocoena*) [170], blue-spotted ribbontail ray (*Taeniura lymma*) [171] and reptile panther chameleon (*Furcifer pardalis*) [172].

Interestingly, *B. microti* positivity was evidenced in mandibular lymph nodes from apparently healthy foxes (*V. vulpes*) assumed to have been contaminated by rodent predation [168].

In 2017, a *B. microti*-like strain was identified in internal organs (heart, lung, spleen, kidney, liver, and reproductive organs) and sometimes in hind limb muscles of marsh frogs in a French farm producing frogs for human consumption [11]. The human pathogenicity of amphibian strains has not been formally demonstrated but cannot be ruled out because the pathogenicity of *B. microti* in wild rodents has been confirmed experimentally in a mouse infection model, with high replication rates in murine macrophages [173].

In cetaceans, *B. ceti* has also been recovered from mandibular, pulmonary, mesenteric, and gastric lymph nodes, spleen, liver, joints, urinary system, and other organs. Noteworthily, Brucellae have been isolated from the female reproductive system, mammary glands, milk and placenta, and in multiple fetal organs, resembling the pathology of terrestrial animals [3,174]. However, it is not the case for pinnipeds (*Pinnipedia*), where most of the isolates came from the spleen, liver, lungs from healthy animals with non-associated pathology. The risk of contamination due to direct contact between coast animals or due to occupational exposure or direct contact with infected aquatic mammals or fomites for humans increases [174].

In both groups, marine mammal parasites, such as lungworms (*Parafilaroides* spp., *Otostrongylus circumlitus*, and *Pseudalius inflexus*), can serve as vectors of marine Brucellae [170,175,176]. In pinnipeds, lungworms are shed in the feces of an infected marine mammal host into the water, then eaten by coprophagic fish; the worms then migrate from the host gastrointestinal tract to their lungs [177], supporting the maintenance and distribution of *Brucella* in aquatic reservoirs.

Recently, *B. pinnipedialis* was recovered for at least 28 days from experimentally infected Atlantic codfish (*Gadus morhua*), suggesting fish as new potential bacterial reservoirs for *Brucella* spp. [178]. Most intriguingly, *B. melitensis* (biovar 3) has been isolated from experimentally and naturally infected Nile catfish (*Clarias gariepinus*) in the delta region of the Nile in Egypt [179]. These findings raise a concern about the role fish and possibly invertebrates may have in transmission or as reservoirs of infection in aquatic environments and for humans, who ingest or handle raw seafood.

*B. inopinata* was originally described in human infections [12,180]. With the current isolation of *B. inopinata*-like (B13-0095) strain from Pac-Man frogs (*Ceratophyrus ornate*) [181] and the first human case of brucellosis caused by an isolate whose genome is identical, the role of wildlife reservoirs should not be underestimated [182].

Considering that multiple *Brucella* species circulate in totally different hosts and environments in wildlife, control and eradication strategies that had succeeded for livestock contexts require adaptation to wildlife reservoir conditions. For instance, Alpine ibexes and goats after experimental conjunctival vaccination with *B. melitensis* Rev.1 vaccine strain display differences in tissue localization and shedding of the bacteria, as well as humoral immune responses [183]. Likewise, retrospective analysis of the effect of vaccination of elks (*Cervus canadensis*), attending winter feed grounds and adjacent areas of western Wyoming, USA, with the S19 vaccine revealed a failure at reducing post-vaccination seroprevalence of *B. abortus* [184]. Vaccination in wildlife reservoirs involves additional challenges to face, as vaccines need to be validated for safety and efficacy in wild animals. Delivery route and cost being also important issues, its application is by far more complicated than that for livestock [185].

Epidemiological surveillance of brucellosis in terrestrial and marine wildlife reservoirs takes place via several methodologies, amongst which serology is favored [186]. In terrestrial wild animals, if serology is the most commonly applied diagnostic approach, PCR appears to be the most sensitive (36.62% of positive results). Isolation from blood samples and visceral organs constitutes the great majority of specimens used for the detection of *Brucella* spp., noting again lymph nodes as a highly prevalent reservoir (94.6%) [187]. Panorama for surveillance in wildlife is challenging given the diversity of laboratory tests, animal species, environments, cross-reactivity, and non-validation of tests for wildlife [188]. This results in uncertain estimates by serological means of the true prevalence for brucellosis in wildlife, requiring cautious interpretations and other technics when available.

Wildlife reservoirs raise major issues in brucellosis. It is clear that carrying or shedding *Brucella* by wild animals bring a potential risk associated with transmission, persistence, and control in this population as well as domestic ones. However, it is still unknown if wildlife hosts are preferential hosts and if wildlife infections represent a critical reservoir of *Brucella* strains for livestock and therefore humans. New results are needed to elucidate the infection cell cycle of *Brucella* in cells of wildlife and better understand the pathological traits of infection related to disease or persistence.

## 6. Conclusions

Thanks to its stealthy nature, upon infection, *Brucella* after entering inside a cell, reaches its intracellular niche, the ER, to replicate sheltered from detection by the immune system. This process is central to *Brucella* as it gives the bacterium the ability to maintain replicating-surviving cycles for long periods of time, even at low bacterial numbers, in its cellular niches. Eventually, *Brucella* will take advantage of the environment provided by its anatomic reservoirs, where the cellular niches reside, to disseminate to other organs, where high replication rates can occur. Of course, an organ reservoir would not exist without a pre-existing intracellular niche. It is also generally well interconnected with the other organs of the host to facilitate dissemination. Secretions and products of natural hosts of *Brucella*, livestock, and wildlife, contribute to contamination spreading. Figure 1 illustrates the journey of *Brucella* inside its hosts and recapitulates the intracellular replicative niche, cellular niches, organ reservoirs, and various hosts described in this report. In this challenging time where the world has seen the rapid emergence of a new viral zoonosis transmitted most probably from a pangolin, it is essential to better understand how another zoonosis, such as brucellosis, develops in its numerous wildlife hosts including bats, and livestock ones. Unraveling the molecular and cellular bases of *Brucella* host preference and reservoirs should be continued to preclude opportunities for *Brucella* to jump hosts.

Moreover, the persistence of viable furtive bacteria for extended periods of time highlights the ability of *Brucella* to maintain a chronic state, a feature that complicates brucellosis treatment, control, and eradication programs. A deeper understanding of the different organ reservoirs of *Brucella* should help to design new therapies, which would overcome the inability of current treatments to reach this surreptitious bacterium in certain cells and organs, as is the case for the bone marrow of infected patients.

*Brucella* niche is distributed in different anatomic reservoirs in the host and especially in some organs, such as the adipose tissue, the role of which in brucellosis is still speculative. This opens up new avenues of research that will undoubtedly contribute to a deeper knowledge of brucellosis and more generally of the mechanisms leading to the chronicity of intracellular pathogens.

## Figures and Tables

**Figure 1 pathogens-10-00186-f001:**
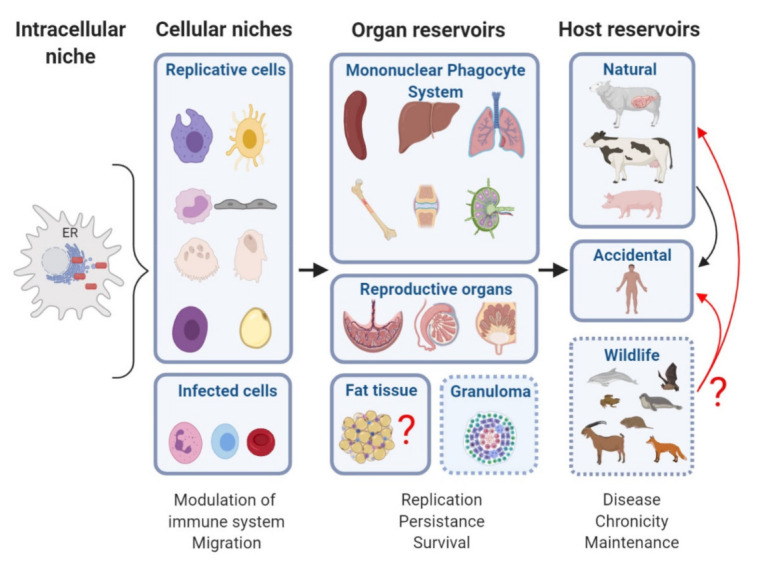
Summary of *Brucella*’s cellular niches and reservoirs. The endoplasmic reticulum is the preferred intracellular niche for *Brucella*, but in some extravillous HLA-G+ trophoblasts, *B. abortus* and *B. suis* are located in lysosomal membrane-associated protein 1 (LAMP1)- and CD63-positive acidic inclusions. *Brucella* replicates in macrophages, dendritic cells, monocytes, trophoblasts, bone cells (osteoclasts, osteoblasts), granulocyte progenitors, adipocytes, and infects other cells such as neutrophils, lymphocytes, and erythrocytes. Some infected cells like neutrophils mediate *Brucella*´s immune response modulation and/or serve as a Trojan horse to disseminate and infect new organs. Several anatomical compartments are populated by *Brucella* infected cells. Organs with high replication rates (placenta, epididymis, mammary glands, lymph nodes, spleen, liver, lungs, and bone marrow) correlate with clinical manifestations of the disease. Once an adaptive immune response is achieved or granulomas contain the infection, *Brucella* develops chronicity and persists at low replication rates. An organ reservoir would not exist without a pre-existing intracellular niche. The structures and physiological characteristics of organ reservoirs allow *Brucella* to start new infection cycles within natural or accidental hosts. Although *Brucella* detection in wild sheep, goats, frogs, fox, bats, and rodents seems almost inconsequential, bacterial loads might be maintained within the host. The zoonotic potential of wildlife reservoirs is still unknown but represents an important risk of transmission to livestock or humans (Created with BioRender.com (accessed on 15 December 2020)).

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
