# Peer review of "Brucella: Reservoirs and Niches in Animals and Humans"

_pathogens, 2021, doi:10.3390/pathogens10020186_

Round 1

Reviewer 1 Report

Dear Authors,

The topic of the review ''Brucella: Reservoirs and Niches in Animals and Humans'' is well explained with scientific context. In my opinion, it is well written and comprehensively described.

Thank you, 

Author Response

We thank Reviewer #1 for his kind acknowledgment of the quality and breadth of our review.

Reviewer 2 Report

Although there is no original contribution nor any news added to the field of Brucella tropism, the present review offers a clear and deep analysis on how Brucella niche is distributed in different anatomic reservoirs in its hosts. Therefore, this could be of help to a profounder knowledge of intracellular pathogens operating mechanisms.

No need to improve English language.  

Author Response

We appreciate the comments of Reviewer #2, who found that our "review offers a clear and deep analysis on how Brucella niche is distributed in different anatomic reservoirs in its hosts". However, we disagree on his first sentence stating that "there is no original contribution nor any news added to the field of Brucella tropism". In contrast, we'd like to emphasize that this review will be the first ever to evoke fat tissue as a potential organ reservoir for Brucella, whose role in brucellosis remains to be demonstrated. We indeed included the only two papers published in the field: The first one published in 2018 ( de Souza et al, ., 2018, Scientific Reports; Reference 149 of our review), "which  described the presence of Brucella in fat cells of the gastro-splenic ligament, next to lipid droplets and precisely where ER is located, in naturally infected fetuses and neonates" and the second one published in 2020 (Viglietti et al., Frontiers in endocrinology Reference 150 of our review), which showed that "B.  abortus replicates in a murine fibroblastic derived adipocyte-like cell line, the 3T3-L1 cells, and in its differentiated adipocyte derivative, albeit with less efficiency". Moreover, we elaborate in this review the potential benefits Brucella might gain from such a reservoir. In addition, the abstract has been modified to support the reviewer’s comment on originality.

Reviewer 3 Report

In this review, entitled: Brucella: Reservoirs and Niches in Animals and Humans, the authors give insights to the current knowledge of the distribution of Brucella within their hosts and they discuss the reservoir potential of wildlife and livestock species. I think the authors gave a comprehensive summary of the niches and reservoir organs of Brucella in this work. I found this manuscript well-written, the structure of this work is clear and logical and grammatically correct. I did not find any major flaws, I only have some minor comment regarding some parts, which I detail below. Overall, I support the publication of this work in the journal Pathogens.

My only comment is regarding the part 5. Reservoirs in wildlife. Sometimes the scientific name is given after the common name and in other cases it is missing, so I suggest adding it everywhere, when the species is being mentioned for the first time. Furthermore, I think this part of the manuscript should be more detailed as there are several other papers on this topic (see some examples below). There are several additional aspects of wildlife and the presence of Brucella could be discussed, including vaccination, prevention, surveillance and distribution of Brucella in wildlife species. 

Bai, Ying, et al. "Molecular survey of bacterial zoonotic agents in bats from the country of Georgia (Caucasus)." PLoS One 12.1 (2017): e0171175.

Davis, D. S., and P. H. Elzer. "Brucella vaccines in wildlife." Veterinary microbiology 90.1-4 (2002): 533-544.

Hernández-Mora, G., J. D. Palacios-Alfaro, and R. González-Barrientos. "Wildlife reservoirs of brucellosis: Brucella in aquatic environments." Revue scientifique et technique (International Office of Epizootics) 32.1 (2013): 89-103.

Batista, Talita Gomes da Silva, et al. "Serologic screening for smooth Brucella sp. in wild animals in Brazil." Journal of wildlife diseases 55.3 (2019): 721-723.

Hernández-Mora, Gabriela, et al. "Brucellosis in mammals of Costa Rica: an epidemiological survey." PLoS One 12.8 (2017): e0182644.

Author Response

We are pleased to see that Reviewer #3 highlighted the up-to-date insight of our review on the reservoirs and niches in Brucella in animals and humans. We thank him for his suggestions as regards the part 5 on "Reservoirs in wildlife", which undoubtedly will improve our manuscript.

As suggested by Reviewer #3, we have homogenized in the whole part the names of wildlife species with the species name first, followed by the scientific name.

We have also extended this paragraph, with additional data and comments on aquatic life, surveillance and distribution of Brucella as well as vaccination (lines 504-506; 529-550; 557-577). However, we did not go too much into details of vaccination and prevention, which are clearly out of the scope of this review and would need a review by themselves. All these changes as well as references added (but Hernandez-Mora et al, 2017, PLoS One, ref #3 already included in the first version of our manuscript) are highlighted in yellow in the highlighted version of our revised manuscript. We also modified the abstract to highlight wildlife reservoirs.